# (20R)-Panaxadiol as a Natural Active Component with Anti-Obesity Effects on ob/ob Mice via Modulating the Gut Microbiota

**DOI:** 10.3390/molecules27082502

**Published:** 2022-04-13

**Authors:** Yuqian Lv, Yining Zhang, Jianshu Feng, Tianyu Zhao, Jingtong Zhao, Yue Ge, Xuehan Yang, Hao Han, Ming Zhang, Li Chen, Mingzhu Xu, Fengying Guan

**Affiliations:** 1Department of Pharmacology, School of Basic Medical Sciences, Jilin University, Changchun 130021, China; lvyq19@mails.jlu.edu.cn (Y.L.); fengjs20@mails.jlu.edu.cn (J.F.); zhaoty19@mails.jlu.edu.cn (T.Z.); zhaojt19@mails.jlu.edu.cn (J.Z.); gey19@mails.jlu.edu.cn (Y.G.); yangxh19@mails.jlu.edu.cn (X.Y.); hanh19@mails.jlu.edu.cn (H.H.); zhangming99@jlu.edu.cn (M.Z.); chenl@jlu.edu.cn (L.C.); 2Department of Pediatric Endocrinology, The First Clinical Hospital Affiliated to Jilin University, Changchun 130021, China; yining@jlu.edu.cn; 3Department of Nephrology, China-Japan Union Hospital of Jilin University, Changchun 130012, China

**Keywords:** (20R)-Panaxadiol, obesity, gut microbiota

## Abstract

Obesity is an important cause of diseases such as type 2 diabetes, non-alcoholic fatty liver and atherosclerosis. The use of ingredients extracted from traditional Chinese medicine for weight loss is now receiving more and more attention. Ginseng has been recorded since ancient times for the treatment of diabetes. The (20R)-Panaxadiol (PD) belongs to the ginseng diol type compounds, which are moderately bioavailable and may remain in the intestinal tract for a longer period of time. This study investigated the potential positive effect of PD in ob/ob mice and evaluated its effect against obesity. The ob/ob mice were administered PD for ten weeks. Our study showed that PD could improve obesity, glucose tolerance disorder, as well as gut dysbiosis. Panaxadiol decreased ob/ob mice’s Firmicutes/Bacteroidetes (F/B). Furthermore, 16S rRNA gene sequencing of the fecal microbiota suggested that PD changed the composition of the gut microbiota in ob/ob mice and modulated specific bacteria such as lactobacillus, prevotellace and so on. Moreover, PD improved the intestinal wall integrity. In conclusion, our results suggest that (20R)-Panaxadiol, as an active ingredient of the traditional Chinese medicinal herb ginseng, may improve obesity to some extent via improving gut microbiota

## 1. Introduction

With the improvement of people’s living standards, industrialized food production and transportation technology has brought a large number of cheap, high calorie, low nutrition food for people’s daily consumption. This is a major factor in the rise of the prevalence of obesity to global epidemic levels. Obesity is an accumulation of fat that affects a pathological state of health. It can lead to a range of diseases such as type 2 diabetes, alcoholic fatty liver disease and atherosclerosis. Obesity and metabolic syndrome are the result of the interaction between the gut microbiota, host genome and diet [1]. Recent studies have confirmed that gut microbiota is associated with human health and disease [2], and that obesity may be related to changes in composition of the gut microbiota [3].

In the midst of complex environmental and genetic factors, gut microbes play a crucial role in regulating obesity and obesity-related metabolic diseases [4,5,6]. It is recognized that diet has approximately 57% influence on the gut microbiota structure, while genetic factors have approximately 12% [7]. As a “virtual metabolic and endocrine organ”, the gut microbiota digests or metabolizes ingested substances into absorbable and sometimes bioactive molecules [8,9]. Accumulating evidence indicates that the etiology of obesity is closely associated with microbiota dysbiosis via inducing a series of metabolites or microbial products in humans and rodents. Therefore, understanding the structure of intestinal flora and its changes may provide new targets and ideas for the treatment of obesity.

In recent years, many natural compounds have been extracted from plants, and their role in the prevention and treatment of obesity and obesity-related metabolic diseases has attracted increasing attention [10,11]. Chinese herbal medicine has shown a powerful capacity for improving both obesity and its metabolic diseases through the regulation of the gut microbiota [12]. Studies have shown that the anti-obesity effect of Ganoderma lucidum is played by regulating intestinal flora [13,14]. As a traditional Chinese medicine, ginseng has been proved to be effective in strengthening the body for thousands of years. It has also been recorded that ginseng has been used to treat diabetes mellitus since ancient times. In recent years, studies have shown that ginseng has a certain anti-obesity effect. The derivative (20R)-Panaxadiol has moderate bioavailability and may persist in the intestinal tract for a longer time. Some studies have shown that Panaxadiol has a certain effect on the treatment of colon cancer and other diseases [15], but the research on its anti-obesity effect is still unclear. In this study, we aimed to investigate the effect of Panaxadiol on obesity and to explore whether its effect is related to intestinal flora.

## 2. Materials and Methods

### 2.1. Materials

The PD used in this study was donated by Professor Chen Yanping (College of Chemistry, Jilin University). Its purity was determined to be more than 98.9% by normalization method of HPLC. Total cholesterol (TC), triglyceride (TG) and total cholesterol (TCHO) diagnostic test kits were purchased from Nanjing Jiancheng Bioengineering Institute (Nanjing, China). Blood glucose test strips were purchased from Roche (Basel, Switzerland). All other reagents were purchased from Beijing Chemical Factory (Beijing, China).

### 2.2. Animals and Treatments

The animal study designed in this paper was approved by the Medical Ethics Committee of Jilin University. A number of C57BL/6J male mice and ob/ob male mice (3 weeks) were purchased from Beijing HFK Bioscience Co., Ltd. (SCXK2019-0008) (Beijing, China). The mice were housed individually in a temperature-controlled animal room with a constant 12 h light/dark cycle. After one week of adaptive feeding, C57BL/6J mice were taken as control (CONTROL *n* = 5), and the ob/ob mice were randomly divided into two groups, ob/ob group (OB *n* = 5), Panaxadiol group (PD *n* = 5). Body weights of mice were measured weekly and treatments were administered by gavage (10 mg/kg, controls were given equivalent amounts of sodium carboxymethylcellulose) from 9:00 to 10:00 am daily. Mice were subjected to OGTT at the seventh week of administration. Mice were anesthetized with isoflurane by inhalation (1.5%) and euthanasia by CO_2_ inhalation in the eighth week. Blood sampling was taken from the medial canthus vein and serum was collected after centrifugation at 3000× *g* (4 °C) for 15 min. Then serum TC, TG, TCHO were determined, according to the manufacturer’s instructions. Samples of subcutaneous adipose tissue (SWAT) and interscapular brown adipose tissue (BAT) were extracted for weighing.

### 2.3. OGTT

For the OGTT, mice were fasted for 12 h and then orally gavaged with glucose dissolved in water at 2 g/kg body weight. Ten microliters of blood were obtained from the tail tip, and the concentration of glucose was measured at 0, 30, 60, 90, and 120 min.

### 2.4. Hematoxylin and Eosin (HE) and In Situ Hybridization (ISH) Stainings

The adipose tissue was quickly removed from the mice, washed with normal saline, dried and then fixed with 10% formalin. After rapid removal of the intestine, it was placed in 4% paraformaldehyde. It was then dehydrated in an ascending series of ethanol, and equilibrated with xylene, followed by embedding in paraffin and sectioning into 5–10 μm slices. Then, the samples were dewaxed with xylene and a descending series of ethanol. Continued sections were stained with both Mayer’s hematoxylin and eosin (HE). The samples were dewaxed, protease treated, denatured, hybridized with designed probes (5′-CY3-GCACGCTACTTGGCTGG-3′) and washed in PBS, then visualization was made using a dark field fluorescence microscope. Intestinal flora in situ hybridization results were manipulated by Servicebio.

### 2.5. Microbiota Sequencing and Data Analysis

Mice feces were collected under aseptic manipulation before anesthesia in the tenth week. The fecal samples were sent to NOVOGENE (Tianjin, China, NovoGene.com) (accessed on 20 September 2021). for 16S rRNA gene sequencing. Total genome DNA from samples was extracted using the CTAB method and sequenced after detection, purification and amplification. Sequences’ analysis were performed by Uparse software (Uparse v7.0.1001 http://drive5.com/uparse/) (accessed on 30 September 2021). Sequences with ≥97% similarity were assigned to the same OTUs, taxonomic information was obtained and the community composition of each sample was counted separately at each taxonomic level: kingdom, phylum, class, order, family, genus and species. Then the samples were homogenized for the next analysis.

### 2.6. Statistical Analysis

All data are expressed as mean ± standard error of the mean. Statistical significance of differences was analyzed by one-way ANOVA with Dunnett test using GraphPad Prism8. *p* < 0.05 was considered to indicate a statistically significant difference.

OTUs’ abundance information was normalized using a standard of sequence numbers corresponding to the sample with the least sequences. Cluster analysis was preceded by principal component analysis (PCA), which was applied to reduce the dimension of the original variables using the FactoMineR package and ggplot2 package in R software (Version 2.15.3). Principal Coordinate Analysis (PCoA) was performed to get principal coordinates and to visualize from complex, multidimensional data. A distance matrix of samples obtained before was transformed to a new set of orthogonal axes, by which the maximum variation factor was demonstrated by the first principal coordinate, and the second maximum one by the second principal coordinate, and so on. PCoA analysis was displayed by WGCNA package, stat packages and ggplot2 package in R software (Version 2.15.3).

## 3. Result

### 3.1. PD Improved Body Weight and Abnormal Glucolipid Metabolism in ob/ob Mice

As shown in Figure 1A–C, PD significantly reduced body weight in mice, starting from the third week after PD administration. Moreover, PD decreased the Lee index and the fat/weight index after 10 weeks. As shown in Figure 1D–F, the serum levels of TG, TC and LDL were higher in the mice of the ob/ob group than those in the control group. Treatment of obese mice with PD significantly reversed the alterations of serum lipid profile. It was showed that PD significantly reduced triglyceride and cholesterol levels. OGTT showed that PD improved insulin sensitivity in ob/ob mice (Figure 1G,H). These results suggest that PD has the ability to improve glucose and lipid metabolism, and can exert a certain degree of anti-obesity effect in ob/ob mice.

### 3.2. PD Improveed Fat Accumulation and Enhanced Intestinal Barrier Integrity in ob/ob Mice

As shown in Figure 2A,B, the HE results showed that subcutaneous inguinal fat (SWAT) as well as dorsal brown fat (BAT) lipid droplets became smaller and showed multi-compartmentality after PD treatment. As shown in Figure 2D,E, the diameter of adipocytes was significantly reduced, indicating that PD made some improvements on the morphology of adipose tissue. The intestine is at the front line of nutrient and lipid absorption, and the integrity of the intestinal barrier is thought to be strongly associated with obesity and other metabolic diseases [16]. As H and E staining (Figure 3C) showed, PD significantly increased the integrity of the intestinal barrier. Scoring of intestinal inflammation in mice, shown in Figure 2F,G, shows that ob/ob mice have mild intestinal inflammation, which is relieved to some extent by PD.

### 3.3. PD Modulated the Overall Structure of the Gut Microbiota in ob/ob Mice

To investigate the effect of PD on the gut microbiota, we sequenced the V3-V4 region of the 16S rRNA gene. Firstly, as shown in Figure 3A, the sparse curve analysis reached a stable level, indicating that the sequencing depth has covered the rare new phylotypes and most of the diversity. PCA analysis showed the overall microbiota structure for each group (PCoA was assessed to compare the overall microbiota structure for each group). The intestinal flora of ob/ob mice were clearly separated from the control group, while PD treatment inhibited the segregation and demonstrated a marked shift close to those in the control group (Figure 3B,C). These results showed that PD could modulate the composition of the gut microbiota.

### 3.4. PD Regulateed the Gut Microbiota of ob/ob Mice at the Phylum, Family and Species Levels

The relative abundance of intestinal flora in each group is shown in Figure 4A. Firmicutes and Bacteroidetes are listed as the two most dominant phylum in each group. It is known that Firmicutes/Bacteroidetes can be used as a featured sign of obesity to some extent. ob/ob significantly increased the ratio of F/B, ob/ob group had significantly increased relative abundance of Firmicutes but reduced relative abundance of Bacteroidetes. Its effects could be conversed by PD (Figure 4B). As the heatmap analysis showed, PD intervention increased the abundance of beneficial bacteria at the phylum, family and species levels, such as Bacteroides, Muribaculaceae, Lactobacillus and Akkermansia (Figure 4A,C,D). As expected, PD could regulate intestinal flora.

### 3.5. Key Phylotypes Responding to the PD Treatment in ob/ob Mice

We also performed LEfSe analysis to identify the specific bacteria that were characteristic among the three groups, discriminative features were identified with LDA score > 4.0. As shown in Figure 5A,B, after gavage with PD, the intestinal flora of the PD group were markedly enhanced in Prevotellacese, Lactobacillus, Alloprevotella and Lachnospiraceae-UCG-006. As shown in Figure 5C, in situ hybridization of Prevotella flora showed a significant increase in the intestine of PD mice compared to ob/ob mice.

Therefore, it can be considered that the beneficial effect of PD on ob/ob mice is related to an enriched abundance of Prevotellacese. For Lactobacillus, the in situ hybridization results did not differ significantly (results not shown). The results showed that PD administration modulated the key phylotypes of gut microbiota by regulating intestinal flora.

## 4. Discussion

Ginseng is one of the most valuable traditional Chinese herbs in China, which has been used for thousands of years as a medicinal herb strengthening the body [11,17]. Studies have shown that ginseng has anti-obesity effects [18]. Recent studies have shown that ginseng, such as Salvia miltiorrhiza, plays an important role in the treatment of obesity [19]. Panaxadiol is the active ingredient of ginseng, which is slightly above 30% bioavailable and can be retained in the intestine for a longer period of time. In this study, the anti-obesity effect of PD was investigated by detecting intestinal microorganisms in ob/ob mice after administration of PD. The results include the following: (1) PD could improve body weight and abnormal glucolipid metabolism in ob/ob mice; (2) PD modulated the overall structure of the gut microbiota in ob/ob mice, and decreased the ratio of F/B; (3) PD changed the composition of the gut microbiota in ob/ob mice and modulated specific bacteria.

After PD administration, the composition of the intestinal flora of mice was changed, the abundance of some beneficial flora, including Muribaculaceae, Akkermansia, Alloprevotella, Lactobacillus and Prevotella, was increased. Muribaculaceae is a member of the Bacteroidales. It has been shown that the weight loss effect of Pu-erh tea is related to the increase in Muribaculaceae flora [20]. In addition, the weight loss and lipid-lowering effect of Chen Pi is also related to Muribaculaceae [21]. Akkermansia, which is usually abundant in healthy individuals, has been shown to be important in maintaining intestinal integrity and attenuating high-fat diet-induced obesity and metabolic syndrome [22,23,24]. Studies provide evidence of a negative association between Akkermansia muciniphila abundance and overweight, obesity, untreated type 2 diabetes or hypertension [25,26,27]. Akkermansia muciniphila prevents obesity and inflammation in HFD-fed mice by improving the intestinal barrier integrity and altering adipose tissue metabolism [28]. Alloprevotella is a bacterium associated with the production of short-chain fatty acids (SCFAS). It has been shown that high-fat diet-induced obese mice with SCFA-producing bacteria (Alloprevotella and Allobaculum), along with increases in fecal SCFA concentration, achieved some weight loss after administration of certain pharmacological stimuli [29]. Lactobacillus is a class of bacteria that can break down sugars for energy and produce large amounts of lactic acid, a probiotic. Lactic acid, produced by Lactobacillus, is a high-quality neurotransmitter that increases satiety after a meal. Lactobacillus rhamnosus activates AMPK, which has a preventive effect in hepatic steatosis and damage [30]. Unfortunately, we did not obtain satisfactory results for Lactobacillus in the in situ hybridization. This may be due to the fact that the obesity of the mice had not yet reached the stage of severe hepatic steatosis or the dose of the drug administered to the mice was not enough. Prevotella is a complex genus. Previous studies have suggested that Prevotella may exacerbate insulin resistance prior to the development of diabetes [31], but more recently it has been suggested that Prevotella can improve glucose metabolism in mice [32], and it has also been shown that the healthy overweight adults with a high abundance of Prevotella lost more fat than subjects with a low abundance of Prevotella, after consuming a whole grain and fiber rich casual diet for six weeks [33]. In addition, Prevotella excels in the extraction of short-chain fatty acid and propionic acid from arabinoxylan and oligofructose. One study showed that chicken eaters had increased fecal Prevotella and increased short-chain fatty acids [34]. Short-chain fatty acids can promote energy expenditure and reduce fat accumulation. This is further evidence that Prevotella may act as a beneficial bacterium in the intestine to exert some anti-obesity effects. Moreover, it was reported that the Prevotellaceae family was found to be abundant in pectin or whole grain oats, which improved the insulin sensitivity and plasma cholesterol profile in previous animal studies [35,36,37]. In this study, Prevotellaceae in situ hybridization was performed in mouse intestinal tissues, and the results showed that Prevotellaceae increased after PD administration. This is in agreement with some previous studies reported and predicted. The ob/ob mice cause genotypic obesity in mice. It has been reported that in the WAT of obese mice, many macrophage-specific or -enriched genes, such as MCP-1, MAC-1, F4/80, and CD68, are dramatically upregulated [38]. It is suggested that the macrophage inflammatory response in WAT is a general phenomenon associated with obesity and that PD may alleviate it. Scoring of intestinal inflammation in mice shows that ob/ob mice have mild intestinal inflammation, which is relieved to some extent by administration of the drug. It is possible that PD would prevent the subsequent increase in obesity and intestinal inflammation in ob/ob mice.

It was shown that PD has an inhibitory effect on weight gain in ob/ob mice. In addition, PD significantly reduced the levels of TCHO and TG in the serum of mice. It may be associated with changes in the composition of the intestinal flora of mice. This study indicated that PD, as a natural active ingredient, can regulate the intestinal microbiota to treat obesity and its complications. Subsequent analysis of serum, as well as fecal metabolites, can be performed in mice, and the mechanism of the anti-obesity effect of each flora in PD can be further explored through fecal transplantation and flora transplantation.

## Figures and Tables

**Figure 1 molecules-27-02502-f001:**
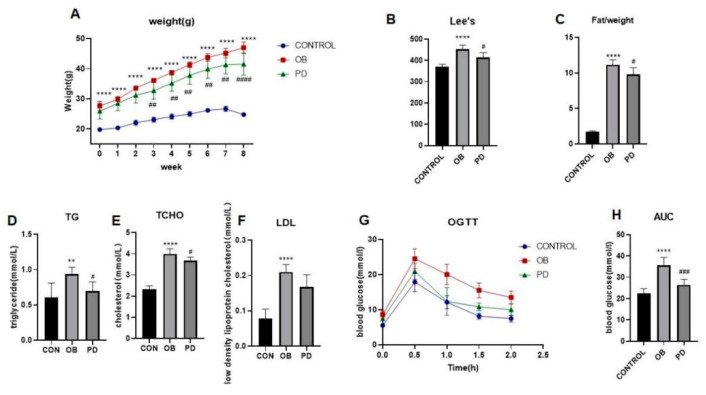
PD improved body weight and abnormal glucolipid metabolism in ob/ob mice. (**A**–**C**) The changes of body weight, Lee’s index and fat/weight (g) during the experimental course; (**D**–**F**) Serum lipid profile including circulating TG, TCHO and LDL levels; (**G**) OGTT after 7 weeks of PD treatment; (**H**) AUC of each group was calculated during the oral glucose tolerance test. ** *p* < 0.01, **** *p* < 0.0001, compared with the CONTROL group. ^#^
*p* < 0.05, ^##^
*p* < 0.01 ^###^
*p* < 0.001, ^####^
*p* < 0.0001 compared with the OB group. *n* = 5.

**Figure 2 molecules-27-02502-f002:**
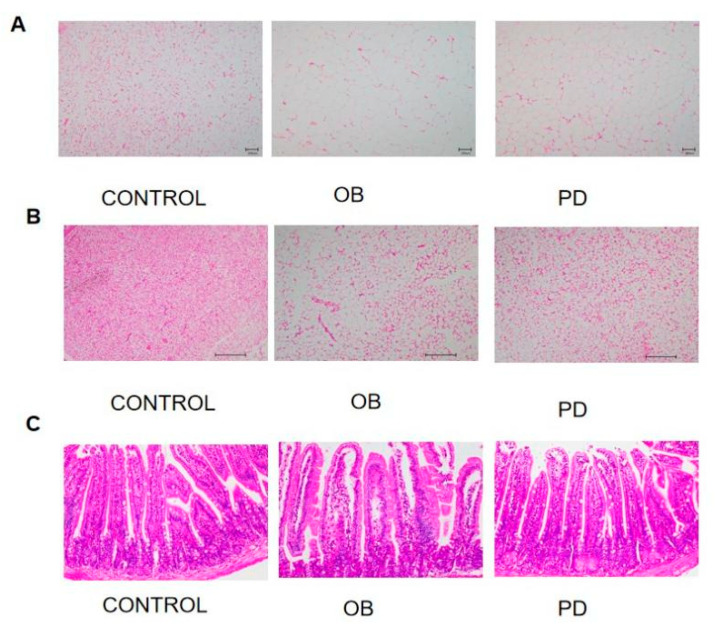
PD improved fat accumulation and enhanced intestinal barrier integrity in ob/ob mice (200×). PD alleviated morphology changes of (**A**) SWAT and (**B**) BWAT in ob/ob mice; (**C**) PD ameliorated intestinal barrier integrity in ob/ob mice. Representative images of HE staining; (**D**) SWAT cell diameter; (**E**) BAT cell diameter. Chart indicating scoring criteria for blinded examination of H and E-stained sections from the intestinal of mice (**G**) and Histopathology score (**F**). **** *p* < 0.0001, compared with the CONTROL group. ^###^
*p* < 0.001, ^####^
*p* < 0.0001 compared with the OB group.

**Figure 3 molecules-27-02502-f003:**
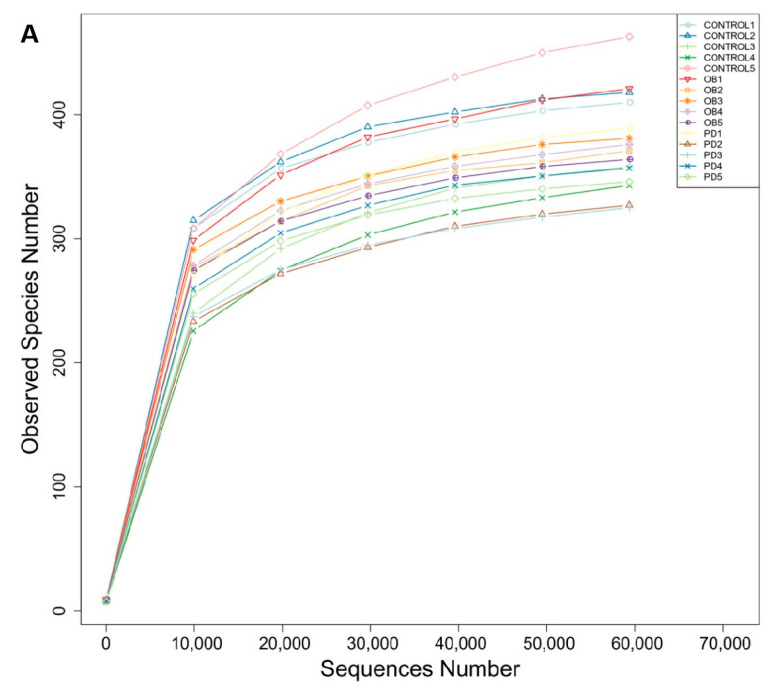
PD modulated the overall Structure of the gut microbiota in ob/ob mice (**A**) Rarefaction curve analysis. The abscissa is the number of sequencings randomly extracted from a sample, and the ordinate is the number of OTUs that can be constructed based on the number of sequencing numbers; (**B**) Plots were generated using the principal coordinates analysis (PCoA); (**C**) principal component analysis (PCA). PCA and PCOA are both based on the OTU level. *n* = 5.

**Figure 4 molecules-27-02502-f004:**
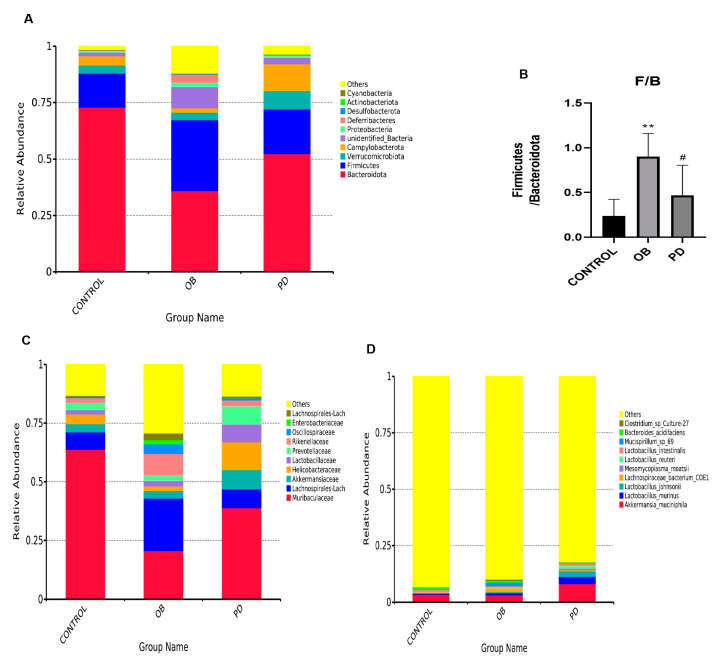
PD regulated the gut microbiota of ob/ob mice at the phylum, family and species levels (**A**) The relative abundance of the 10 top-ranked phyla were presented; (**B**) The ratio of Firmicutes/Bacteroidetes; (**C**,**D**) The relative abundance of the 10 top-ranked families and species were presented. ** *p* < 0.01, compared with the CONTROL group. ^#^
*p* < 0.05, compared with the OB group. *n* = 5.

**Figure 5 molecules-27-02502-f005:**
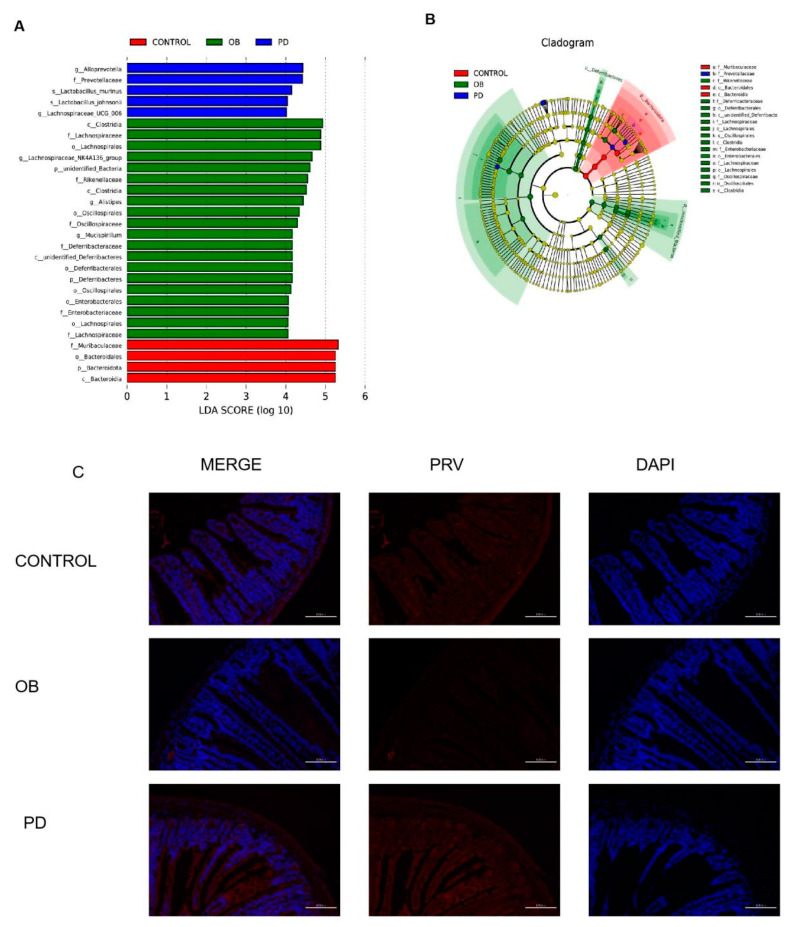
Key Phylotypes Responding to the PD Treatment in ob/ob Mice (**A**) Linear discriminant analysis (LDA) scores were computed for taxa with differential abundance in the fecal microbiota of CONTROL, OB and PD. The LDA score indicated the effect size and ranking of each differentially abundant taxon (LDA > 4); (**B**) Taxonomy cladogram. The circle of radiation from inside to outside represented the taxonomic rank from phylum to genus (or species) and the diameter of the circles was based on relative abundance; (**C**) In situ hybridization (ISH) staining (200×) (Prevotella).

## Data Availability

The data generated or analyzed during the study are included in the article.

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
