# Peer review of "(20R)-Panaxadiol as a Natural Active Component with Anti-Obesity Effects on ob/ob Mice via Modulating the Gut Microbiota"

_molecules, 2022, doi:10.3390/molecules27082502_

Round 1

Reviewer 1 Report

In this study the anti-obesity effect of PD was investigated in ob/ob mice. The authors showed that after administration was able to improve body weight and abnormal glucolipid metabolism. Furthermore, it was suggested that the microflora is completely altered after the administration of PD and it can regulate the intestinal microbiota to treat obesity and its complications, including hyperglycemia. 

From my point of view this manuscript is redy for publication without further revisions. It is well writen, clear, and well organized. 

Author Response

Many thanks for your suggestions, we have revised the articles, such as 168, 179, 227, 233.

Reviewer 2 Report

The work of Lv et al, shows the potential benefit of PD in obesity through the change of gut microbiota and improvement of intestinal wall integrity.

The experiments presented are well-conducted. Nevertheless, some analyses are not deep enough to rise some conclusions. Specifically, the conclusions stated in figure 2.

The sentence “lipid droplet became smaller…” (line 147) is conclude after qualitative analysis. I suggest to analyze quantitative by measuring lipid droplet size to make the result more robust. In addition, the authors can include some analysis of adipose tissue inflammation such as the evaluation of crown-like structures.

Moreover, the increase in the integrity of intestinal barrier is again merely observational. I suggest to evaluate deeper by using some scoring measurement (see this paper as an example, DOI:10.3390/ijms20122907).

Therefore, further analysis should be performed to explore better the potential beneficial effect of PD.

Author Response

  1. The experiments presented are well-conducted. Nevertheless, some analyses are not deep enough to rise some conclusions. Specifically, the conclusions stated in figure 2.The sentence “lipid droplet became smaller…” (line 147) is conclude after qualitative analysis. I suggest to analyze quantitative by measuring lipid droplet size to make the result more robust. In addition, the authors can include some analysis of adipose tissue inflammation such as the evaluation of crown-like structures.

Thank you for your suggestion, for the quantification of adipocyte diameter we have addressed in FIG2D,E, described in line 153-155. For adipose inflammation we refer to some articles and give additional information in line 274-278.

  1. Moreover, the increase in the integrity of intestinal barrier is again merely observational. I suggest to evaluate deeper by using some scoring measurement (see this paper as an example, DOI:10.3390/ijms20122907).Therefore, further analysis should be performed to explore better the potential beneficial effect of PD..

The intestinal inflammation scores were supplemented in FIG2F,G and illustrated in line 158-160 and 278-282. Thank you very much for the recommended article.

Round 2

Reviewer 2 Report

The authors have addressed all my comments/suggestions.